# Color-Tunable White LEDs with Single Chip Realized through Phosphor Pattern and Thermal-Modulating Optical Film

**DOI:** 10.3390/mi12040421

**Published:** 2021-04-12

**Authors:** Zhenpeng Su, Bo Zhao, Zheng Gong, Yang Peng, Fan Bai, Huai Zheng, Sang Woo Joo

**Affiliations:** 1Institute of Technological Sciences, Wuhan University, Wuhan 430072, China; zhenpeng_su@whu.edu.cn; 2School of Power and Mechanical Engineering, Wuhan University, Wuhan 430072, China; bozhao2019@whu.edu.cn (B.Z.); zheng_g@whu.edu.cn (Z.G.); 3School of Aerospace Engineering, Huazhong University of Science and Technology, Wuhan 430074, China; ypeng@hust.edu.cn; 4School of Mechanical Engineering, Yeungnam University, Gyeongsan 712-749, Korea; baifan@ynu.ac.kr

**Keywords:** light-emitting diodes (LEDs), correlated color temperature (CCT), LED packaging, paraffin-PDMS film

## Abstract

In this paper, a new method to regulate the correlated color temperature (CCT) of white light-emitting diodes (LEDs) is proposed for the single-chip packaging structure, in which the blue light distribution emitted from the chip in the red/yellow phosphor layer was modulated through changing the paraffin-polydimethylsiloxane (PDMS) film transparence and haze. The results show that the transmittance of the paraffin-PDMS film can be modulated from 49.76% to 97.64%, while the haze of that ranges from 88.19% to 63.10%. When the thickness of paraffin-PDMS film is 0.6 mm, and the paraffin-PDMS film concentration is 30 wt%, the CCT of white LED decreases from 15177 K to 3615 K with the increase of thermal load in the paraffin-PDMS film. The modulating range of its CCT reaches 11562 K. The maximum CCT variation at the same test condition is only 536 K in the repeated experiments within one week.

## 1. Introduction

White light-emitting diodes (LEDs) have been extensively used in traffic lights, backlight displays, display screens, automobile lighting, indoor lighting, and many other fields [1,2,3,4]. Currently, most white LEDs show the fixed correlated color temperature (CCT). The CCT of white LEDs has a profound impact on daily human activities, including comfort, health [5,6], and thinking [7,8]. For example, in the retail and service industries, adjusting the CCT of LEDs can make the environment more attractive to customers. According to time, music, mood, etc., changing the CCT in real-time can better highlight the stage visual effects in an evening party. Besides, the CCT of natural light changes in real-time every day. The change of CCT with the time that simulates natural light can result in a positive visual experience. Therefore, real-time adjustment of CCT is a significant trend in the development of LEDs [9,10].

The CCT of white LEDs can be adjusted in two ways: the spectra of the LED chip, or that of the phosphor. The nanorod and nanowire technology to adjust LED light color at the chip level has been extensively studied, and the phosphor-free white LEDs have been realized [11,12,13,14]. Meanwhile, the CCT of white light is tuned by blending an appropriate fraction of the as-synthesized different color emitting nanocrystals [15,16]. Most of these methods, however, cannot adjust the CCT in real-time.

To realize the real-time adjustment of the CCT, several methods have been proposed. The simplest way is to arrange two or three different color LED chips and adjust the CCT by changing the mixing ratio of different lights, but it requires a complicated control circuit, and is also restricted by uniformity of the color spaces [17,18,19,20,21]. Quantum dot materials are used to adjust the CCT, such as changing the volume ratio of different color quantum dot materials and introducing a dynamic color filter. However, the heat generated during chip operation will result in a decrease in the luminous efficiency and light conversion efficiency of the quantum dot material after being heated [22,23,24]. Moreover, the liquid crystal (LC) technology is combined with LED to realize CCT adjustment through a color conversion film, but the adjustable range of CCT is limited to only 2500 K [25,26]. Therefore, a simple, wide range and real-time CCT regulation method of LEDs is still needed.

For the present research, a new method of regulating CCT of white LEDs with a single chip was developed through phosphor pattern and thermal-modulating optical film. This method is based on modulating the blue light radiation distribution on the patterned red/yellow phosphor layer through changing the paraffin-polydimethylsiloxane (PDMS) film transparence and haze by the thermal stimulation. In this study, the influence of paraffin-PDMS film on the light path and spatial intensity distribution of blue LED chip under different thermal stimulations were analyzed, and the mechanism of CCT regulation was obtained. The manufacturing process of paraffin-PDMS film was introduced. The transmittance and haze of paraffin-PDMS film under the thermal stimulation were tested, and the related optical properties of single-chip packaged LED were also tested. It analyzed the effect of CCT adjustment by combining the phosphor pattern and thermal modulating optical film. The stability and practicability of this method were examined.

## 2. Principles and Experiments

Figure 1 showed the schematic diagram of the CCT-modulation white LED module and its work principle. The white LED with tunable CCT with single-chip was mainly composed of an indium tin oxide (ITO) conductive glass, paraffin-PDMS film, and a patterned phosphor layer. The peak wavelength of the blue LED chip was 460–470 nm and the chip was fixed on an aluminum substrate. An optical lens was mounted above the chip for regulating the blue light emitted from the chip to transmit within 15° angular angles, as shown in Figure 1b. The paraffin-PDMS film was placed on the ITO glass. The distance between the ITO glass and the LED chip was 12 mm. The patterned phosphor layer in which the red phosphor was with the diameter of 12 mm and yellow phosphor had the inner diameter of 12 mm, and the outer diameter of 32 mm was placed above the paraffine-PDMS film. And their gap was set as 2 mm.

In our experiments, the mixture of the red phosphor and the silicone was fabricated firstly. Then the mixture of yellow phosphor and silicone was manufactured after the red phosphor was cured. The phosphor pattern with a thickness of 350 μm was fabricated by the screen printing process on a high transmittance glass plate [27]. The concentrations of red phosphor mixture and yellow phosphor mixture were 40 wt% and 20 wt% respectively. The red phosphor had the peak photoluminescence wavelength of 640 nm, while that of the yellow phosphor was 564 nm.

As shown in Figure 1, the light emitted by the blue LED passed through a lens, ITO conductive glass, paraffin-PDMS film, a red/yellow phosphor layer, and finally irradiated in the air. The ITO was etched by a specific pattern with two lines with 1.5 mm indicated by the shape line shown in Figure 1c. When the current crossed the ITO film, there was uniform Joule heat on the ITO glass. The center was heated rapidly due to the large resistance in the central area. The paraffin-PDMS film became transparent when it was heated [28,29], as shown in Figure 1f. The central heating of the conductive glass caused the paraffin in the film to melt from approximately the center circle. The supply voltage of ITO conductive glass determined the size of the paraffin melting area, which affected the transmittance and haze. The increase of transmittance and decrease of haze in the center of the film was due to the melting of the central paraffin. Hence, the blue light passing through the center of the film in Figure 1g was more concentrated than that in Figure 1e, resulting in more red phosphors was excited. By changing the voltage of conductive glass, the melting area of the film was regulated. Then the different spatial intensity distribution of blue LED was controlled to change the excitation power ratio of red/yellow phosphor, and realize the real-time adjustment of the CCT. The different CCTs brought people different visual experiences, which could be seen from the CIE (Commission Internationale de l’Eclairage) chromaticity diagram of Figure 1h.

As shown in Figure 2a, the manufacturing process of paraffin-PDMS film was introduced. Paraffin is a white solid at room temperature and melts into a transparent liquid at 58 °C [30,31]. In the paraffin-PDMS film, the paraffin particles were screened using 60 mesh and 150 mesh gauze with diameters between 100–250 μm. Thus its size was less than 250 μm. The thickness of the paraffin-PDMS film was 0.8 mm. In order to control the film thickness, a plastic ring dam with a height of 0.8 mm and an inner diameter of 33 mm was designed. In experiments, the dam was placed on the ITO glass surfaces. The paraffin-PDMS mixture was filled into the dam. The mixture volume was adjusted to guarantee the same thickness between the mixture and the dam. Due to the low melting temperature of paraffin, we chose the curing temperature at 40 °C. The paraffin-PDMS film with a thickness of 0.8 mm was obtained after 48 h of curing in a drying oven. After curing, the dam was removed from the paraffin-PDMS film. The film had the properties of paraffin heating transparency and PDMS polymer softness. Figure 2b showed three concentrations (10 wt%, 20 wt%, 30 wt%) of the paraffin-PDMS film with a film thickness of 0.8 mm. We could see that the concentration of paraffin affected the transmittance of the film.

## 3. Results and Discussion

To explore the connections between the central temperature and the supply voltage of paraffin-PDMS film, the ITO conductive glass with a particular pattern was applied voltage, and an infrared thermal imager was used to measure the temperature. Figure 3a showed the infrared thermal images of the film surfaces under different voltages, and the heat transfer was approximately circular. The temperature in the center of the film was the highest, and the farther away from the center, the lower the temperature. The square resistance of ITO conductive glass was less than 15 Ω/cm^2^, the film thickness was 135 nm, and the transmittance was more than 94.5%. The paraffin concentrations of different films were 10 wt%, 20 wt%, 30 wt%, and the film thickness was 0.6 mm. The melting point of paraffin was 58 °C. From the variation curve of film center temperature with voltage shown in Figure 3b, it could be seen that under the same voltage, the difference in film center temperature of different concentrations was small, and as the voltage increased, the film center temperature raised. As shown in Equation (1), a quadratic polynomial equation was used to fit the center temperature curve of the film with a concentration of 30 wt%.
(1)$$T=28.175+0.576u+2.519u^{2}$$

The correlation coefficient *R*^2^ could reach 0.9996. According to the fitting formula, the relationship between the film center temperature and the supply voltage could be clearly obtained.

As shown in Figure 4a, the melting area of the paraffin-PDMS film was approximately circular and positively correlated with the voltage. To conveniently test the transmittance and haze in the central area of the film, the through-hole diameter of the transmittance haze tester was adjusted to 10 mm. For comparison, the transmittance and haze of the 0.6 mm paraffin layer were measured. When the paraffin was melted, the transmittance could reach 100%, and the haze could be as low as 1.05%. However, at room temperature, the transmittance was only 61.32%, and the haze could get 88.51%. As shown in Figure 4b, the transmittance and haze of paraffin-PDMS films with different concentrations changed with voltage, and the film thickness was also 0.6 mm. For films with different concentrations, when the voltage was between 0–2.4 V, the temperature did not reach the melting point of paraffin, and the transmittance and haze were almost unchanged. When the voltage was between 2.6–4.4 V, the temperature reached the melting point of paraffin, as the central paraffin continuously melted, the transmittance increased rapidly, while the haze decreased rapidly. When the voltage reached 4.4 V, the diameter of the melting area was larger than 10 mm, reaching the maximum diameter of the test through-hole, and the transmittance and haze tended to be stable. The transmittance range of the film with a paraffin concentration of 30 wt% was the largest, from 49.76% to 97.64%, and the haze range was smaller, from 88.96% to 83.77%. When the paraffin concentration of the film was 10 wt%, the transmittance changed from 68.50% to 95.60%, and the haze decreased from 88.19% to 63.10%. According to the test data of transmittance and haze of paraffin-PDMS film, it could be seen that after the paraffin was melted, the transmittance of the film with different concentrations had little difference, but the transmittance value at room temperature was very different, and the test data of haze was just the opposite. The results showed that under the same thickness the main factor that affected the transmittance and haze of the film was the concentration of paraffin.

To explore the influence of different paraffin concentrations on CCT, the CCT of white LEDs was measured at the driving voltage of 5 V and the driving current of 500 mA. As shown in Figure 5, the CCT changed slowly, then quickly, and finally tended to be stable, which could be well explained by the variation curve of transmittance and haze. When the voltage was 0–2 V, the transmittance and haze hardly changed, and the CCT was basically unchanged. When the voltage was 2–4 V, the center transmittance of the film increased rapidly, and the haze decreased rapidly, which led to more light emitted from the center to excite the red phosphor, and the CCT decreased rapidly. When the voltage was 4–5 V, as the melting region increased, the excitation ratio of red/yellow phosphors reached equilibrium, and the CCT tended to be stable and reached the lowest. When the thickness of the paraffin-PDMS film was 0.6 mm and the paraffin concentration was 30 wt%, as the voltage increased, the CCT decreased from 15177 K to 3615 K, and the modulating range of CCT reached 11562 K. Figure 5b was the CCT of LED under different voltage. As the voltage decreased, the CCT increased significantly. Figure 5c showed the CIE chromaticity diagram under different voltages of ITO glass, and the change of CCT could be seen more intuitively. The values of CCT, color rendering index (CRI), CIE coordinates were shown in Table 1. It could be seen that the CRI decreased with the CCT. The maximum value was 81.5, while the minimum one was about 66.3. However there was a large CCT modulation range, the light quality should be optimized by choosing the phosphor material and phosphor pattern in future work.

As shown in Figure 6a, the electroluminescence spectra further demonstrated the regulation process of the CCT. As the proportion of red wavelengths increased, the CCT of LED decreased significantly. The LED distribution curve was tested, as shown in Figure 6b. The luminous intensity with the center temperature of 90 °C in the paraffin-PDMS was higher than that at 26 °C and the maximum difference was 0.308 cd. The smaller the absolute value of the angle, the greater the difference of luminous intensity, which was due to the change of transmittance and haze in the center of the film. This method could simulate natural light well. By adjusting the parameters, the LED could achieve high luminous intensity and CCT from 9:00 to 15:00, about 6500 K. At sunrise and sunset, the LED had lower light intensity and CCT, about 3500 K.

To verify the stability of CCT adjustment, the CCT of LED was tested within one week, as shown in Figure 7. The CCT fluctuated in a certain range, and the maximum difference was 536 K at the same voltage of ITO glass. The difference was small, which could meet the requirements of normal operation. This experiment had good stability and reliability for CCT adjustment of white LEDs. The luminous efficiency of the LED calculated by measured lumen value and power consumption of LED chips was 29.08 lm/W at the CCT of 6500 K, and that without the paraffine-PDMS film was 55.32 lm/w. The power to adjust the CCT was analyzed. The maximum voltage was 4.6 V, and the current was 0.11 A. In other words, the maximum power to adjust the CCT was only 0.506 W. Taking it into account the entire power, the LED luminous efficiency was 24.93 lm/W. The efficiency loss was due to the scattering effect of the paraffin particles in the paraffin-PDMS film. The luminous efficiency could be further improved by improving the process, such as changing the diameter of paraffin particles, changing the smoothness of paraffin particles, and changing the thickness of the film. In this works, we mainly demonstrated the principle of CCT modulation. In future work, we will further optimize the process to improve luminous efficiency.

## 4. Conclusions

A new method of changing the CCT of white LEDs was proposed. By changing the melting state of paraffin in PDMS film, adjusting the spatial intensity distribution, and changing the absorption and excitation power ratio of phosphors with different colors, the CCT of White LEDs can be adjusted smoothly from 15177 K to 3615 K, and has the advantage of low energy consumption. In this way, LEDs can simulate natural light and give people a more comfortable visual experience.

## Figures and Tables

**Figure 1 micromachines-12-00421-f001:**
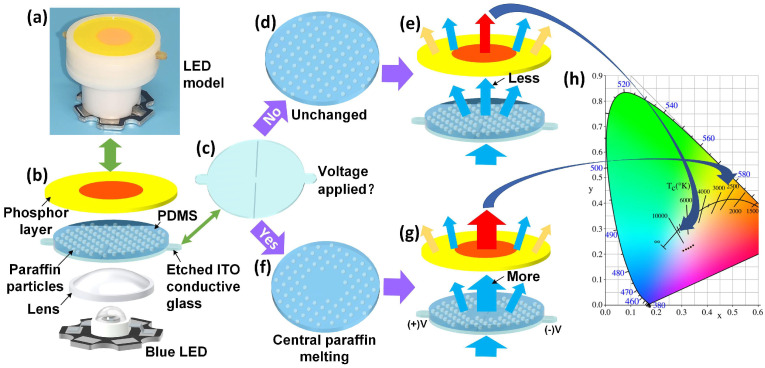
The schematic diagram of the light-emitting diode (LED) module and the correlated color temperature (CCT) adjustment. (**a**) Picture of LED module. (**b**) Schematic diagram of LED module. (**c**) Etched indium tin oxide (ITO) conductive glass. (**d**) Paraffin-polydimethylsiloxane (PDMS) film. (**e**) The spatial intensity distribution. (**f**) Centrally melted paraffin-PDMS film. (**g**) The spatial intensity distribution with the central paraffin melting. (**h**) Commission Internationale de l’Eclairage (CIE) chromaticity diagram.

**Figure 2 micromachines-12-00421-f002:**
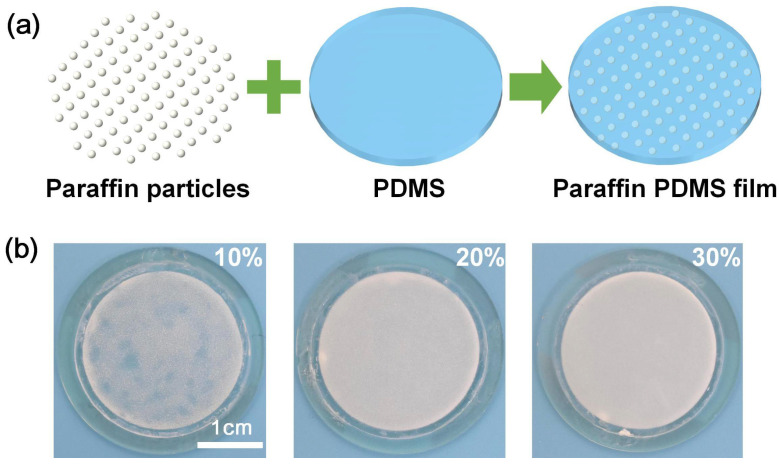
(**a**) Schematic of fabrication of a paraffin-PDMS film. (**b**) Picture of paraffin-PDMS films with concentrations, 10%, 20%, and 30%.

**Figure 3 micromachines-12-00421-f003:**
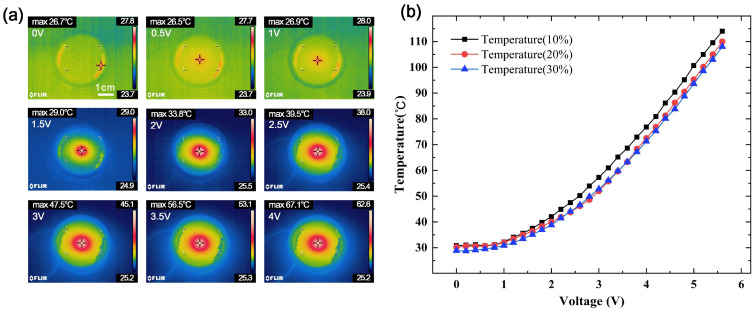
(**a**) Infrared thermal images of film surfaces under different voltages. (**b**) Variation curve of film center temperature with voltage.

**Figure 4 micromachines-12-00421-f004:**
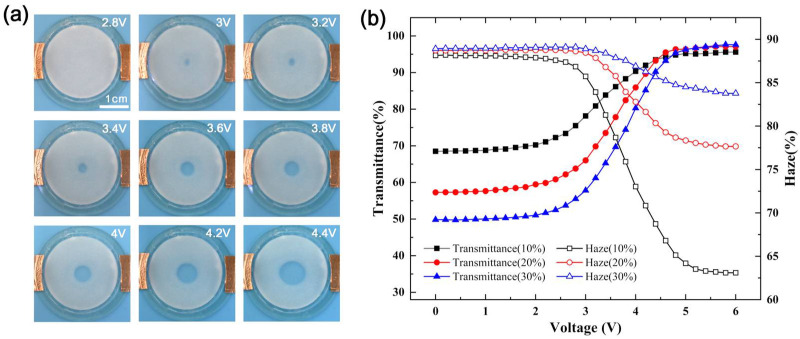
(**a**) Pictures of films with melting centers. (**b**) Transmittance and haze of paraffin-PDMS films with different concentrations changed with voltage.

**Figure 5 micromachines-12-00421-f005:**
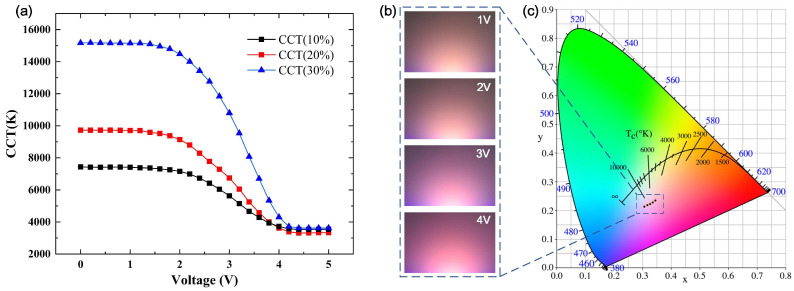
(**a**) Variation curves of CCT VS voltage of ITO glass. (**b**) LED light color under different voltages of ITO glass. (**c**) Chromaticity diagram at different voltages of ITO glass.

**Figure 6 micromachines-12-00421-f006:**
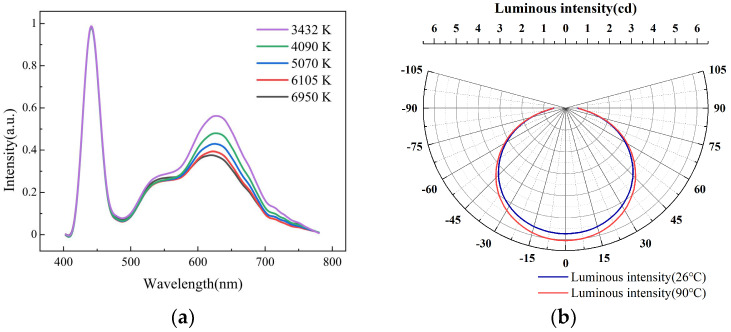
(**a**) Electroluminescence spectra of white lights with different CCTs. (**b**) Light distribution curves of LEDs.

**Figure 7 micromachines-12-00421-f007:**
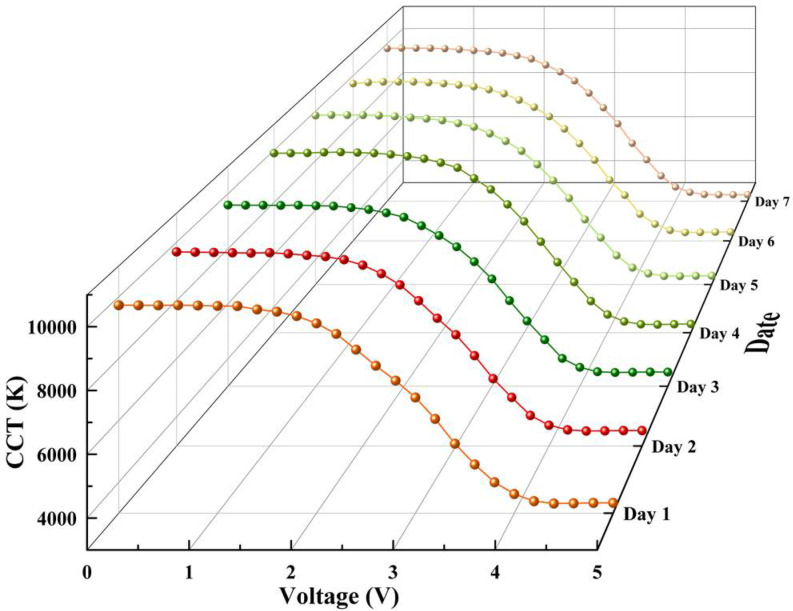
CCT change curves of LEDs VS voltage of ITO glass in one week.

**Table 1 micromachines-12-00421-t001:** The values of correlated color temperature (CCT), color rendering index (CRI), Commission Internationale de l’Eclairage (CIE) coordinates.

CCK(K)	CRI	*x*	*y*
3432	81.50	0.3569	0.2578
4090	76.45	0.3441	0.2464
5070	70.38	0.3349	0.2432
6105	66.65	0.3277	0.2402
6905	66.26	0.3222	0.2426

## Data Availability

The datasets are available from the corresponding author on reasonable request.

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
