# Peer review of "Color-Tunable White LEDs with Single Chip Realized through Phosphor Pattern and Thermal-Modulating Optical Film"

_micromachines, 2021, doi:10.3390/mi12040421_

Round 1
Reviewer 1 Report
In this manuscript the authors proposed a creative way to modulate CCT of white LEDs. A combination of paraffin and PDMS operated at different voltages has been used to manipulate the output intensity and CCT of LEDs. The influence of paraffin-PDMS weight of 10, 20 and 30 % has been investigated. The authors demonstrated a CCT change from 3615K to 15177K. The paper is well organized and contains interesting results. I recommend publication after some minor changes.
- The English needs to be improved. I recommend the authors to rephrase all the 2-3 line sentences.
- The last sentence in the abstract “The presented method shows the high stability with multi experiments” has to be changed, to convey a scientific term.
- The introduction is strong and convincing. Nanostructured LEDs has been proposed by some companies like OSRAM, LG, GLO AB, and Aledia and some research groups to achieve white LEDs and CCT modulation. I recommend the authors to read and if they think it is appropriate to cite the following papers.
- Monemar, Bo, et al. "Nanowire-based visible light emitters, present status and outlook." Semiconductors and Semimetals. Vol. 94. Elsevier, 2016. 227-271.
- Lim, Wantae, et al. "SiO2 nanohole arrays with high aspect ratio for InGaN/GaN nanorod-based phosphor-free white light-emitting-diodes." Journal of Vacuum Science & Technology B, Nanotechnology and Microelectronics: Materials, Processing, Measurement, and Phenomena 34.4 (2016): 042204.
- Nami, Mohsen, et al. "Tailoring the morphology and luminescence of GaN/InGaN core–shell nanowires using bottom-up selective-area epitaxy." Nanotechnology 28.2 (2016): 025202.
- Kishino, K., Yanagihara, A., Ikeda, K. and Yamano, K., 2015. Monolithic integration of four-colour InGaN-based nanocolumn LEDs. Electronics Letters, 51(11), pp.852-854.
- Can authors comment on how they were able to control the PDMS thickness? (PDMS processing steps).
- The caption of figure.1 is divided into two parts pleas fix it.
- 48 hours of cure time is very long, have authored tested higher temperature and a shorter time? (line 103)
- Have authors tested the impact of long-term operation on CCT modulation efficiency?
- The color-scale bar in figure 3 is not readable.
- Authors need to mention that adding any extra electronic part (CCT modulation) into a LED, will reduce the wall plug efficiency. Because, now you need more power to operate a LED in a preferred condition.
- The sentence in line 183 has to be changed. Intense light is not a scientific term.
- In figure 6, do 26 C and 90 C refer to figure 3 (b)? if not, what is 26C and 90C?
- What is the intensity difference between 26 c and 90C? (add a specific number)
- How did authors calculate the luminous efficiency of 29.08 lm/W?
- Did you consider the power have been used to modulate the CCT? This is also part of power consumption that most people ignore.
- What is the operation condition of the LED (L-J-V)?
- 08 lm/w is very low, what is the lm/w for the LED without the paraffine-PDMS? (what is loss?)
- Can authors comment on how all the following aspects can improve the efficiency?
- diameter of paraffin particles,
- changing the smoothness of paraffin particles
- changing the thickness of the film.
- What would be the optimum operation condition?
- Paraffin weight
- Voltage
Author Response
Dear Reviewer,
Please see the attachment.
Sincerely yours,
Huai Zheng
School of Power and Mechanical Engineering
Wuhan University
Wuhan, China 430072
E-Mail: huai_zheng@whu.edu.cn

Reviewer 2 Report
This work focuses on the tunable white light from extreme cool white to warm white by reporting a new pattern of the fabrication of LED. The controlled distribution of different colors for tuning CCT of white light is very common. However, in this work, the contribution of phosphors for white light is controlled (indirectly) by the input voltage is interesting. A single-chip packaging structure of phosphors for generating white light is a significant part of this work. In general, this is an interesting work. The manuscript is well written. I recommend its publication in Micromachines after improving the following details.
- The authors are suggested to provide more details about the yellow and red phosphors.
- In this single-chip structure of two different phosphors, is there any probability of phosphors diffusion between yellow and red phosphors? The authors are suggested to talk briefly in the manuscript.
- In this structure of LED, does the phosphors film touch the paraffin? If touch, what is the effect of temperature on stability or degradation of phosphors? If not, is there any difference in the output with varying the separation?
- The authors are suggested to provide the reason for having low luminous efficacy in the manuscript.
- The authors are suggested to follow some related work: Adv. Opt. Mater. 0, 1900916 (2019); Nanoscale Adv., 2019,1, 1791-1798.
- The authors are suggested to provide the corresponding color rendering index (CRI) of the white light so that the reader can compare the quality of light concerning different CCT.
- In this manuscript, the main concern for the large audience will be the spectra of the white light. How do you calculate the CCT or luminous efficiency of white light? Did you collect any spectra (such as electroluminescence spectra or any other) of white light? If so, the authors are suggested to provide some spectra such as the spectra of extremities (CCT) and some spectra between the extremities.
Author Response
Dear reviewer,
Please see the attachment.
Sincerely yours,
Huai Zheng
School of Power and Mechanical Engineering
Wuhan University
Wuhan, China 430072
E-Mail: huai_zheng@whu.edu.cn

Round 2
Reviewer 1 Report
Dear Authors,
Thank you for your efforts to address all comments.
Well done, and good luck.
Reviewer 2 Report
The manuscript in this form is publishable in the Micromachines without the need for further revision.